# Cell cycle and age-related modulations of mouse chromosome stiffness

**Ning Liu[1], Wenan Qiang[2,3], Philip W Jordan[4,5], John F Marko[6,7]\*, Huanyu Qiao[1]\***

[1]Department of Comparative Biosciences, University of Illinois at Urbana-Champaign, Urbana, United States; [2]The Chemistry of Life Processes Institute, Northwestern University, Evanston, United States; [3]Division of Reproductive Science in Medicine, Department of Obstetrics and Gynecology, Feinberg School of Medicine, Northwestern University, Chicago, United States; [4]Biochemistry and Molecular Biology Departments, Johns Hopkins University Bloomberg School of Public Health, Baltimore, United States; [5]Biochemistry and Molecular Biology Department, School of Medicine, Uniformed Services University of the Health Sciences, Bethesda, United States; [6]Department of Molecular Biosciences, Northwestern University, Evanston, United States; [7]Department of Physics and Astronomy, Northwestern University, Evanston, United States

**\*For correspondence:**
john-marko@northwestern.edu (JFM);
hqiao@illinois.edu (HQ)

## eLife Assessment

This **valuable** paper describes the stiffness of meiotic chromosomes in both oocytes and spermatocytes. The authors identify differences in stiffness between meiosis I and II chromosomes, as well as an age-dependent increase in stiffness in meiosis I (and meiosis II) chromosomes, results that are highly significant for the field of chromosome biology. The report is, however, mostly descriptive and the mechanisms underlying age-dependent changes in chromosome stiffness remain unclear. The evidence suggesting that changes in stiffness are independent of cohesin, which is known to deteriorate with age, is still **incomplete**.

**Abstract** Chromosome structure is complex, and many aspects of chromosome organization are still not understood. Measuring the stiffness of chromosomes offers valuable insight into their structural properties. In this study, we analyzed the stiffness of chromosomes from metaphase I (MI) and metaphase II (MII) oocytes. Our results revealed a tenfold increase in stiffness (Young's modulus) of MI chromosomes compared to somatic chromosomes. Furthermore, the stiffness of MII chromosomes was found to be lower than that of MI chromosomes. We examined the role of meiosis-specific cohesin complexes in regulating chromosome stiffness. Surprisingly, the stiffness of chromosomes from three meiosis-specific cohesin mutants did not significantly differ from that of wild-type chromosomes, indicating that these cohesins may not be primary determinants of chromosome stiffness. Additionally, our findings revealed an age-related increase of chromosome stiffness for MI oocytes. Since aging is associated with elevated levels of DNA damage, we investigated the impact of etoposide-induced DNA damage on chromosome stiffness and found that it led to a reduction in stiffness in MI oocytes. Overall, our study underscores the dynamic and cyclical nature of chromosome stiffness, modulated by both the cell cycle and age-related factors.

## Introduction

DNA serves as the carrier of genomic information, interacting with a wide variety of proteins to form chromatin during interphase. Before cell division, chromatin is compacted into thick, noodle-like chromosome structures, a process which plays a critical role in ensuring the equitable distribution of genetic material (*Vagnarelli, 2013*). This process is essential for maintaining genome integrity, as any errors in chromosome behavior during meiosis or mitosis can lead to conditions such as cancer, infertility, miscarriage, or congenital diseases (*Potapova and Gorbsky, 2017*; *Theisen and Shaffer, 2010*). Despite many decades of research, chromosome structure and organization are not yet understood.

Chromosome compaction during mitosis and meiosis is not a random process, but a tightly regulated sequence of events. In mammals, centimeters-long segments of DNA are compacted into chromosomes less than 10 μm long, via a multi-level process (*Kschonsak and Haering, 2015*). This process begins with the formation of nucleosomes, wherein DNA wraps around histone octamers, following the 'beads on a string' model (*Alberts et al., 2002*). Linker histone H1 binds to the entry and exit points of DNA on the nucleosome and interacts with the linker DNA region between nucleosomes.

However, the interaction between histones and DNA does not come close to explaining the full extent of chromosome compaction, indicating that other factors must be involved in this process (*Wood and Earnshaw, 1990*; *Paulson et al., 2021*; *Batty and Gerlich, 2019*). There are different models for chromosome organization, with the most popular being the 'Scaffold/Radial-Loop' model (*Laemmli et al., 1978*; *Maeshima and Laemmli, 2003*; *Paulson and Laemmli, 1977*; *Earnshaw and Laemmli, 1983*) and the 'Chromatin Network' model (*Biggs et al., 2019*). The Scaffold/Radial-Loop model proposes a continuous central protein core with chromatin loops stacked around it, based on electron microscopy observations. The Chromatin Network model suggests that structure proteins link chromatin segments without forming a continuous central core. MNase, an enzyme that degrades DNA/chromatin, ablated chromosome stiffness, in discord with a pure Scaffold/Radial-Loop model and instead indicating a key role for chromatin cross-bridges (*Poirier and Marko, 2002*). Experiments with DNases, which affect chromosome stiffness depending on their cutting frequencies, lend support to the Chromatin Network model (or equivalently, a chromosome radial looping model without a continuous proteinaceous scaffold and with some degree of crossbridging between adjacent loops).

Chromosome compaction and decompaction occur in different phases of the cell cycle (*Antonin and Neumann, 2016*), and the shape and size of chromosomes vary accordingly (*Zickler and Kleckner, 2023*). It is reasonable to hypothesize that chromosome stiffness also differs between stages, and between mitotic and meiotic chromosomes, which have similar shapes but experience different molecular events (*Marko and Siggia, 1997*; *Koshland and Strunnikov, 1996*). Chromosome stiffness measurements have shown that prophase I spermatocyte chromosomes are approximately 10 times stiffer than those in mitosis, indicating differences in chromosome organization between meiosis I and mitosis (*Biggs et al., 2020*).

During meiosis I, a unique railroad-track-like protein structure, known as the synaptonemal complex (SC), forms between the two homologous chromosomes (*Heyting, 1996*; *Hollingsworth, 2020*). Although the SC was thought to be a potential factor increasing stiffness of meiotic chromosomes, it has been observed that SYCP1, a key SC component, does not contribute to chromosome stiffness (*Biggs et al., 2020*). SYCP1 laterally connects the two 'rails' of the SC, axial elements (AEs), but may not impact the longitudinal stiffness. It is more likely AEs provide mechanical strength to the meiotic chromosomes longitudinally. Cohesin proteins, which connect sister chromatids in both mitosis and meiosis, are fundamental components of AEs (*Peters et al., 2008*; *Ishiguro et al., 2011*). In meiosis, meiosis-specific cohesin proteins, such as SMC1β, RAD21L, REC8, and STAG3, may play a role in chromosome structure and stiffness (*Zheng and Xie, 2019*; *Rong et al., 2016*; *Gyuricza et al., 2016*).

Aging can also induce significant changes in chromosomes, potentially leading to apoptosis, senescence, or cancer, all of which affect the lifespan and well-being of both animals and humans (*Shammas, 2011*; *Faggioli et al., 2011*). Moreover, chromosome-associated protein levels change with age, with some increasing while others decreasing (*Thakur, 1983*; *Oliviero et al., 2022*). For instance, cohesin proteins along chromosome axes decrease with age, potentially contributing to increased rates of aneuploidy and unsuccessful gamete production (*Lister et al., 2010*; *Nakagawa and FitzHarris, 2017*). Given the elevated aneuploidy rates in oocytes from older individuals, it is essential to explore how aging impacts chromosome mechanics and to gain a better understanding of the molecular mechanisms driving these changes.

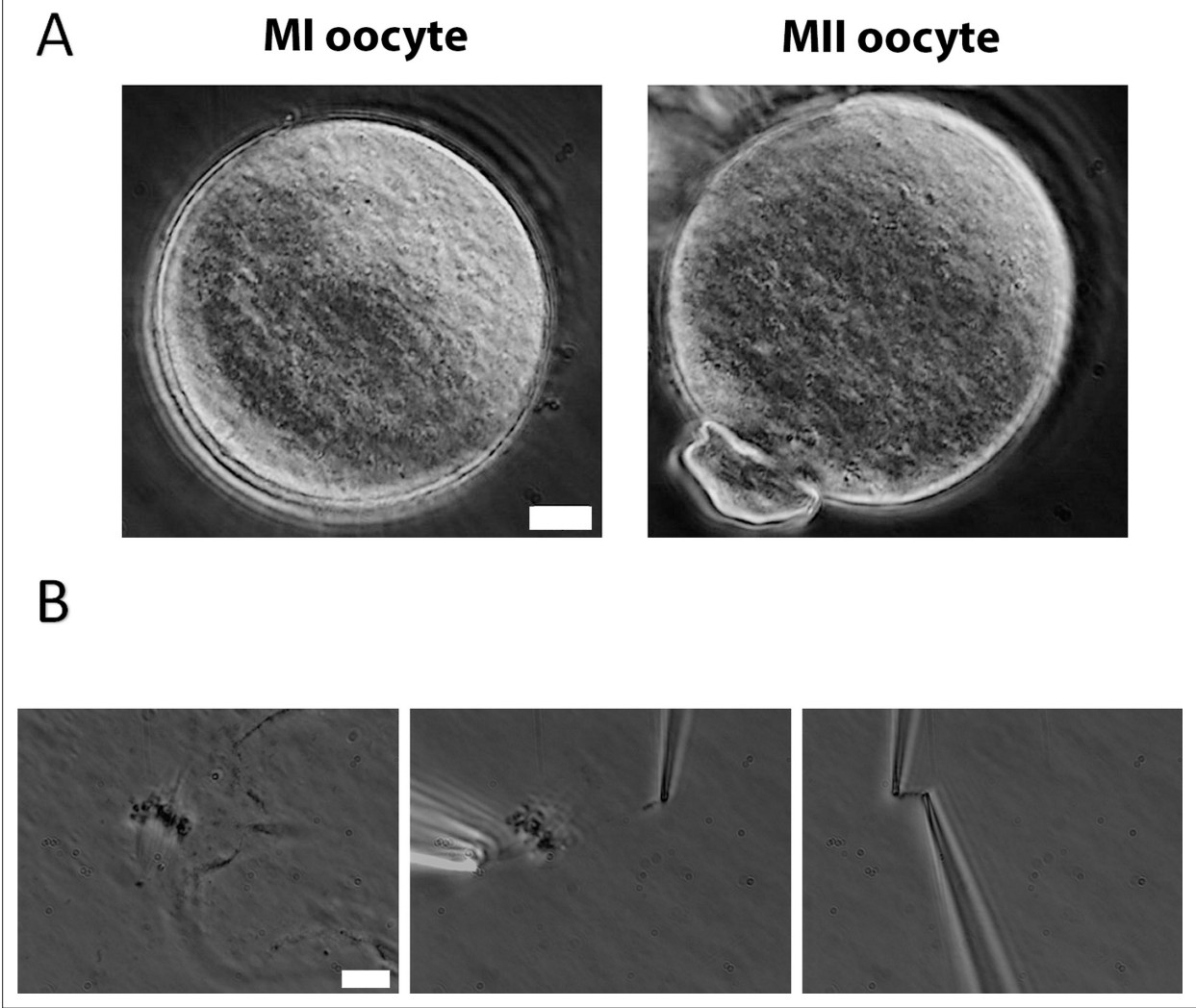

**Figure 1.** Chromosome isolation from oocytes. (**A**) Oocytes after zona pellucida removal. Left panel: metaphase I (MI) oocyte. Right panel: metaphase II (MII) oocyte with visible polar body. Scale bar = 10 μm. (**B**) Spindle isolation process. Left panel: spindle flowing out from the oocyte after oocyte lysis. Middle panel: a chromosome being isolated from the spindle–chromosome complex. Right panel: chromosome captured between two pipettes. Scale bar = 10 μm.

The online version of this article includes the following source data and figure supplement(s) for figure 1:

**Figure supplement 1.** Representative images of oocyte chromosome stretching.

**Figure supplement 1—source data 1.** Related to *Figure 1—figure supplement 1*, left panel.

**Figure supplement 1—source data 2.** Related to *Figure 1—figure supplement 1*, right panel.

**Figure supplement 2.** Chromosome stiffness is stable in phosphate-buffered saline (PBS) solution and resistant to Triton X-100 treatment.

**Figure supplement 2—source data 1.** Related to *Figure 1—figure supplement 2*, left panel.

**Figure supplement 2—source data 2.** Related to *Figure 1—figure supplement 2*, right panel.

## Results

### Chromosome stiffness measurement for metaphase I and metaphase II mouse oocytes

To measure oocyte chromosome stiffness, we isolated chromosomes from oocytes collected from 3- to 4-week-old mice. These oocytes were cultured for 6 hr to reach metaphase I (MI) or 14 hr to reach metaphase II (MII). The zona pellucida was removed by treating the oocytes with Tyrode's solution for approximately 3 min (*Figure 1A*). The oocyte membranes were then lysed via microspraying Triton X-100, allowing the oocyte contents to flow out spontaneously. Using this technique, we successfully

isolated the spindle from the oocytes (*Figure 1B*, left) and subsequently separated the chromosomes from the spindle (*Figure 1B*, middle).

Next, we used two pipettes with small openings to grasp the two ends of the chromosome (*Figure 1B*, right). This setup allowed us to measure chromosome stiffness by moving one pipette and observing and calibrating the bending of the other (see Materials and methods for further details). We stretched and relaxed the chromosomes to monitor its length change under an applied force, ultimately determining their Young's modulus (a measure of material stiffness which is independent of geometry, i.e., chromosome thickness or number of chromatids). These experiments were carried out with extensions of less than twice the chromosome's native length, ensuring a reversible mechanical response—that is, the force versus extension curve was similar during both stretching and retraction (*Figure 1—figure supplement 1*).

## Chromosome stiffness in MI oocytes is about 10 times higher than that in mitotic cells

Chromosome stiffness has been studied for a variety of mitotic cells, revealing similarities and differences across different cell types (*Hornick et al., 2015*; *Sun et al., 2018*; *Strom et al., 2021*). However, a comprehensive analysis of chromosome stiffness through either mitotic or meiotic cell cycles has not been done. For comparison with the meiotic case, we measured the chromosome stiffness of mouse embryonic fibroblasts (MEFs) at late pro-metaphase (just slightly before their attachment to the mitotic spindle) and found that the average Young's modulus was 340 ± 80 Pa (*Figure 2B*). The value is consistent with our previously published data, where the modulus for MEFs was measured to be 370 ± 70 Pa (*Biggs et al., 2020*).

Next, we isolated chromosomes from mouse MI oocytes and measured their stiffness (*Figure 2A*), obtaining a Young's modulus of 3790 ± 700 Pa, roughly tenfold higher than that of MEF chromosomes (*Figure 2B*). This finding was comparable to previous results demonstrating that spermatocyte prophase I chromosomes are approximately 10 times stiffer than MEF chromosomes (*Biggs et al., 2020*). These results suggest that the high chromosome stiffness observed in meiotic cells is a feature of gametes, common to both sexes. To further explore this, we investigated chromosome stiffness in MII oocytes and explored potential factors that might contribute to the high stiffness observed for gamete chromosomes.

## The stiffness of chromosomes in MI mouse oocytes is significantly higher than that of MII oocytes

To study the effect of meiotic cell cycle stage on chromosome stiffness, we measured the chromosome stiffness for the MII oocytes, as we did for the MI chromosomes. We found that chromosome stiffness in MII oocytes was significantly lower than that in MI oocytes: the Young's modulus of MII oocytes was 670 ± 130 Pa, while that of MI oocytes was 3790 ± 700 Pa (p < 0.001; *Figure 2B*). Surprisingly, despite this reduction, the stiffness of MII oocyte chromosomes was still significantly higher than that for mitotic cells (*Figure 2B*). This finding challenges the conventional view that meiosis II is closely analogous to mitosis, since we observe that chromosome mechanics in meiosis II quantitatively differs from that observed in mitotic cells (*Hochwagen, 2008*). Our results affirm that chromosome stiffness varies dynamically across different cell cycle stages.

To verify the consistency of chromosome measurements, we compared our data with previously published results (*Biggs et al., 2020*), in terms of the 'doubling force' (the force required to double the length of a chromosome, which is expected to be dependent on chromosome thickness). MEF chromosomes in the published study exhibited a doubling force of 190 ± 40 pN, while wild-type (WT) prophase I spermatocytes had a doubling force of 2130 ± 440 pN (*Biggs et al., 2020*). Our independent measurements of these quantities closely agreed with the prior results, giving a doubling force of 210 ± 40 pN for mitotic MEFs and 1690 ± 450 pN for WT prophase I spermatocytes (*Figure 2—figure supplement 1*), indicating quantitative reproducibility of the results (notably, different researchers carried out the two sets of experiments). Here, we found that the doubling forces of chromosomes from MI and MII oocytes are 3770 ± 940 and 510 ± 50 pN, respectively. Chromosomes from MI oocytes are much stiffer than those from both mitotic cells and MII oocytes (*Figure 2—figure supplement 1*), in terms of either Young's modulus or doubling force.

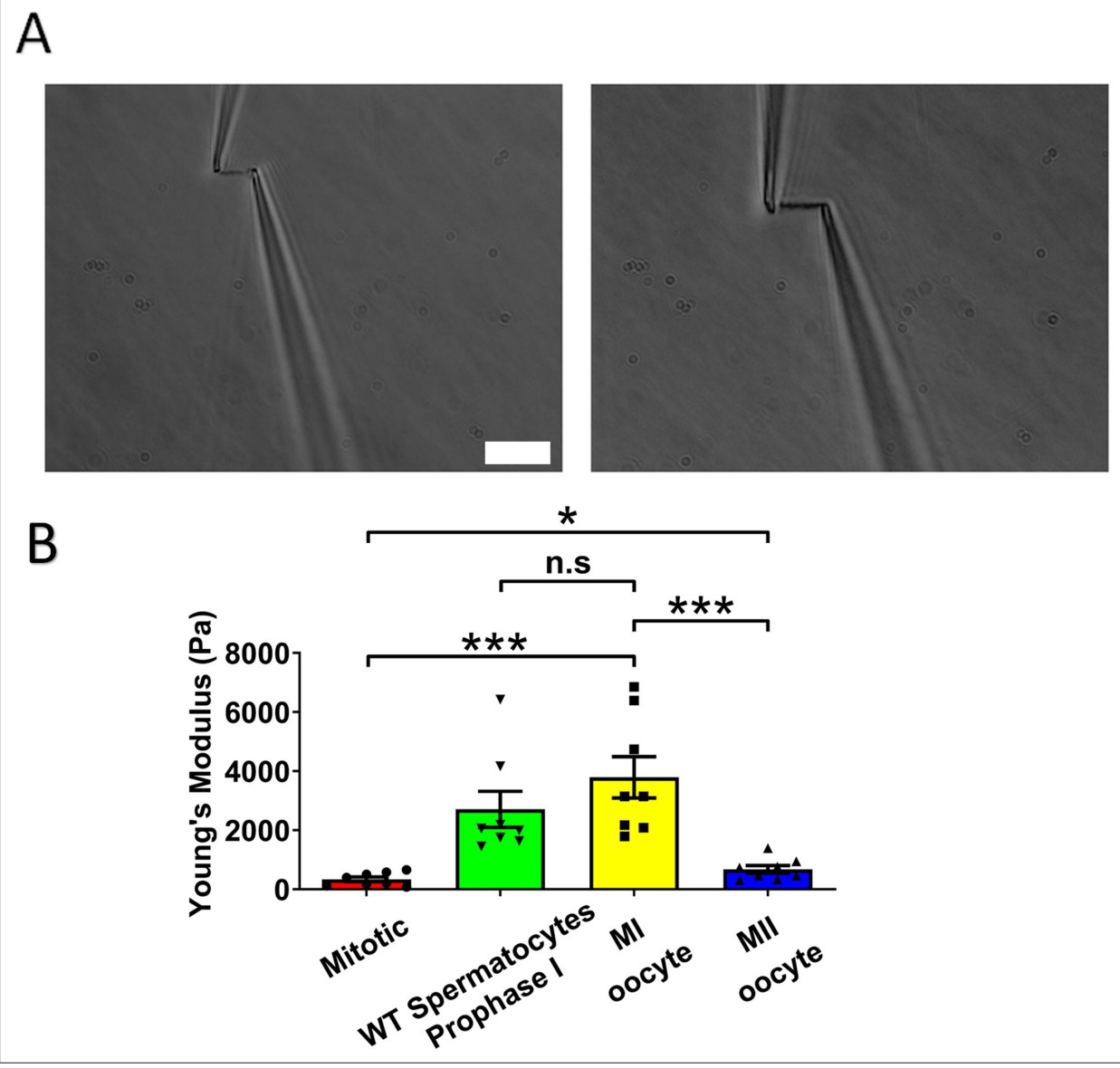

**Figure 2.** Chromosome stiffness measurement. (**A**) Example images of chromosome isolation. Left: metaphase II (MII) oocyte chromosome. Right: metaphase I (MI) oocyte chromosome. Scale bar = 10 μm. (**B**) Chromosome stiffness comparison across different cell types: mitotic cells ($n = 8$)., wild-type (WT) spermatocytes at prophase I ($n = 8$), MI oocytes ($n = 8$), and MII oocytes ($n = 8$). Young's modulus of MI oocyte chromosomes (3790 ± 700 Pa) is much higher than that of mitotic cells (370 ± 70 Pa, p = 0.0002) and MII oocytes (670 ± 130 Pa, p = 0.0006). Data are presented as mean ± SEM. All statistical analyses were performed via $t$-test, n.s, non-significant, ($p > 0.05$), *$p < 0.05$, **$p < 0.01$ and ***$p < 0.001$.

The online version of this article includes the following source data and figure supplement(s) for figure 2:

**Source data 1.** Related to *Figure 2B*.

**Figure supplement 1.** Chromosome doubling force comparison across different types of cells.

**Figure supplement 1—source data 1.** Related to *Figure 2—figure supplement 1*.

## Meiosis-specific cohesins do not contribute to chromosome stiffness

We previously demonstrated that the central elements of the SC do not contribute to longitudinal chromosome stiffness, so we shifted our focus to the role of meiosis-specific cohesins during meiosis I (*Biggs et al., 2020*). Cohesin can load onto chromosomes before SCs form (*de Vries et al., 2005*). During mammalian mitosis, cohesin proteins bind along the chromosome axis during S phase, staying at the centromeres until anaphase, when they are cleaved by separase. Most chromosome-arm cohesion proteins are removed early before metaphase–anaphase transition by a separase-independent

pathway (*McGuinness et al., 2005*). During meiosis I, cohesin proteins are removed from chromosome arms at anaphase I by separase, and only a small amount remains at the centromere until anaphase II (*Lee et al., 2003*). While cohesin proteins disappear from chromosome arms by metaphase in both mitosis and MII, they are retained along chromosome arms during MI.

Given the higher cohesion levels along chromosome arms during MI, we hypothesized that this might be the reason MI oocyte chromosomes have greater stiffness relative to those from MII. To test this, we studied the effects of mutation of meiosis-specific cohesins (REC8, STAG3, and RAD21L) on chromosome stiffness by utilizing $Rec8^{-/-}$, $Stag3^{-/-}$, and $Rad21l^{-/-}$ mutant mice (*Ward et al., 2016*). These cohesins are essential for sister chromatid cohesion, and their absence disrupts gametogenesis, resulting in arrest at prophase I (*Xu et al., 2005*; *Hopkins et al., 2014*; *Herrán et al., 2011*). This arrest forces us to carry out experiments on prophase I chromosomes. Furthermore, we are forced to carry out experiments in males where spermatocytes are continuously produced and are surgically accessible; female oocyte prophase I occurs in utero where isolation and genotyping are not tractable. Nevertheless, we can compare results for WT and mutant spermatocytes.

We isolated chromosomes from $Rec8^{-/-}$ prophase I spermatocytes, which displayed large and round cell size and thick chromosomal threads, indicative of advanced chromosome compaction after stalling at a zygotene-like prophase I stage (*Figure 3B*). The combination of large cell size and degree of chromosome compaction allowed us to reliably identify $Rec8^{-/-}$ prophase I chromosomes. Using micromanipulation, we measured chromosome stiffness by stretching the chromosomes (*Figure 3B*; *Biggs et al., 2019*). Surprisingly, there was no significant difference in chromosome stiffness between wild-type (WT) control and $Rec8^{-/-}$ mutant (2710 ± 610 Pa in WT spermatocytes vs. 2580 ± 620 Pa in $Rec8^{-/-}$ spermatocytes, p = 0.8884) (*Figure 3E*).

Similarly, for both $Stag3^{-/-}$ (2710 ± 610 Pa in WT spermatocytes vs. 2240 ± 210 Pa in $Stag3$ mutant spermatocytes, p = 0.4533) and $Rad21l^{-/-}$ (2710 ± 610 Pa in WT spermatocytes vs. 2050 ± 370 Pa in $Rad21l^{-/-}$ spermatocytes, p = 0.3514) mutants, no significant difference in chromosome stiffness relative to WT was observed (*Figure 3C–E*). We concluded that meiosis-specific cohesins do not play a dominant role in determining chromosome stiffness.

We also note that we compared prophase I chromosome mechanics for WT CD-1 spermatocytes with mutants from a C57BL/6J background. To check whether strain might be a factor, we conducted additional experiments to compare the spermatocyte chromosome stiffness between WT CD-1 and C57BL/6J mice. The results showed no significant difference (2710 ± 610 Pa in CD-1 vs. 3290 ± 610 Pa in C57BL/6J, p = 0.512), suggesting that chromosome stiffness is consistent across strains (*Figure 3— figure supplement 1*).

## Chromosomes from older MI oocytes have higher stiffness than those from younger MI oocytes

We next examined the effects of aging on chromosome mechanics. Initially, we hypothesized that chromosomes from aged oocytes would be less stiff based on previous findings that aging is associated with decreased levels of cohesin, particularly REC8 (*Tian et al., 2021*; *Tsutsumi et al., 2014*). To test this hypothesis, we isolated chromosomes from MI oocytes of 48-week-old mice (nearing the end of fertility, roughly equivalent to 40-year-old humans) and compared them to chromosomes from 3- to 4-week-old mice (*Figure 4A*). Contrary to our hypothesis, our measurement revealed that chromosomes from older mice were much stiffer than those from younger mice (8150 ± 1590 Pa in older MI oocytes vs. 3790 ± 700 Pa in younger MI oocytes, p = 0.0150) (*Figure 4B*). This result further supports the conclusion that cohesins are not the main contributors to chromosome stiffness.

Our findings are consistent with previous observations of increased chromosome stiffness in aged MII oocytes when compared to their counterparts from younger oocytes (*Hornick et al., 2015*). The doubling force for chromosomes in 3- to 4-week-old MII oocytes was measured at 510 ± 50 pN (see *Figure 2—figure supplement 1*), while for 6- to 8-week-old MII oocytes, it was significantly higher at 830 ± 100 pN (*Hornick et al., 2015*). These findings underscore a trend of increased chromosome stiffness with advancing age, common to both MI and MII oocytes.

At the MII stage, most cohesin complexes have already dissociated from chromosome arms, and only a small amount remains, connecting sister chromosomes at their centromeres until anaphase II. Therefore, the observed age-related increase in chromosome stiffness is unlikely to be driven by cohesin levels. This suggests that other age-related factors, possibly linked to chromosome structural

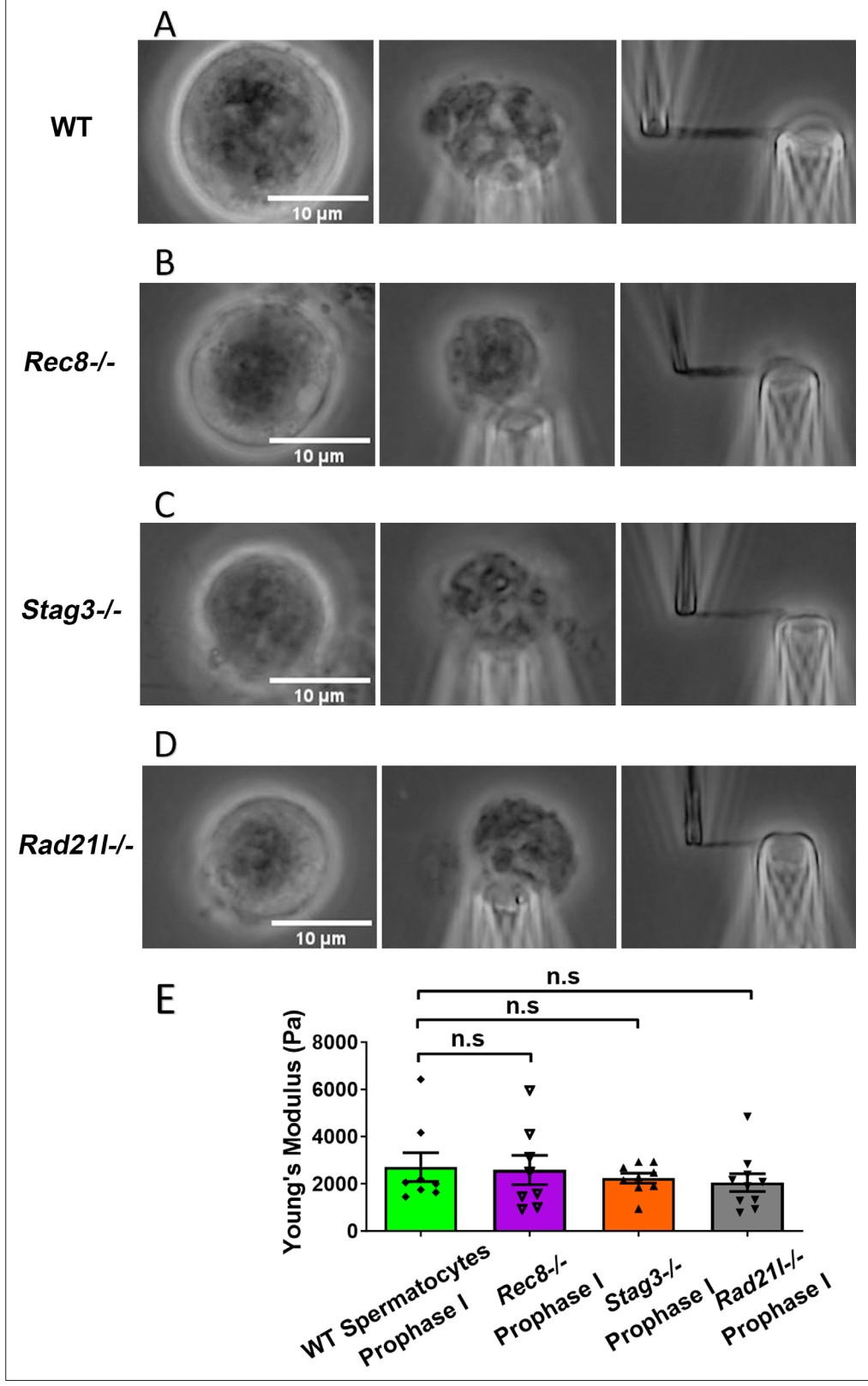

**Figure 3.** Chromosome stiffness measurement in meiosis-specific cohesin mutants. (**A**) Images of chromosome isolation from wild-type (WT) spermatocytes. Scale bar = 10 μm. (**B**) Images of chromosome isolation from *Rec8$^{-/-}$* spermatocytes. Scale bar = 10 μm. (**C**) Images of chromosome isolation from *Stag3$^{-/-}$* spermatocytes. Scale bar = 10 μm. (**D**) Images of chromosome isolation from *Rad21l$^{-/-}$* spermatocytes. Scale bar = 10 μm. (**E**) Chromosome

*Figure 3 continued on next page*

*Figure 3 continued*

stiffness comparison across various cell types: WT spermatocytes at prophase I ($n$ = 8), $Rec8^{-/-}$ spermatocytes at prophase I ($n$ = 8), $Stag3^{-/-}$ spermatocytes at prophase I ($n$ = 9), and $Rad21l^{-/-}$ spermatocytes at prophase I ($n$ = 10). Young's modulus of WT spermatocyte chromosomes (2710 ± 610 Pa) is not significantly different from that of $Rec8^{-/-}$ spermatocyte chromosomes (2580 ± 620 Pa, p = 0.8884), $Stag3^{-/-}$ spermatocyte chromosomes (2240 ± 210 Pa, p = 0.4533), and $Rad21l^{-/-}$ spermatocyte chromosomes (2050 ± 370 Pa, p = 0.3514). Data are presented as mean ± SEM. All statistical analyses were conducted using $t$-test. n.s, non-significant.

The online version of this article includes the following source data and figure supplement(s) for figure 3:

**Source data 1.** Related to *Figure 3E*.

**Figure supplement 1.** Comparison of chromosome stiffness between CD-1 and C57BL/6 mice.

**Figure supplement 1—source data 1.** Related to *Figure 3—figure supplement 1*.

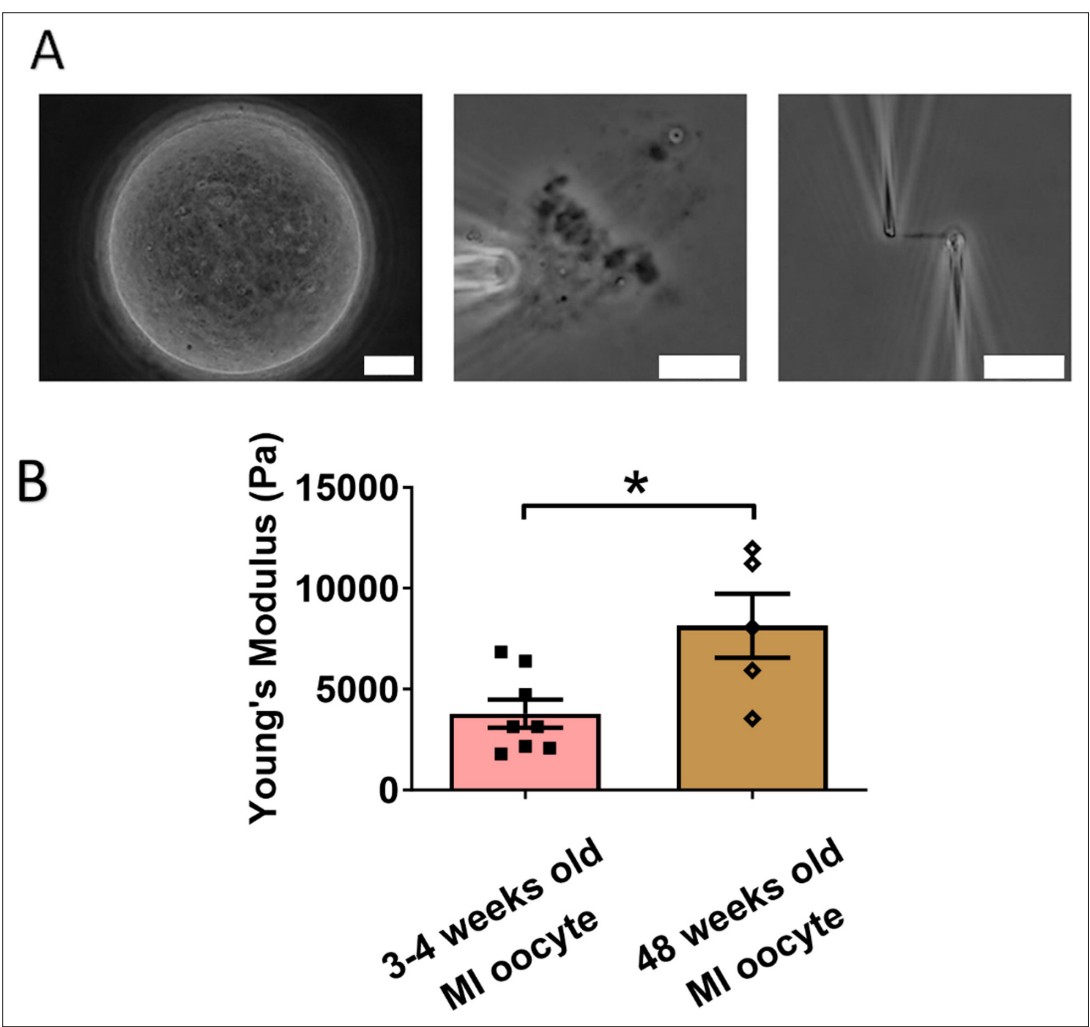

**Figure 4.** Chromosomes in aged oocytes are stiffer than those in younger oocytes. (**A**) Images showing the isolation of chromosomes from an aged metaphase I (MI) oocyte. Left panel: aged MI oocyte images. Middle panel: spindle isolated from aged MI oocyte. Right panel: MI chromosome isolated from the spindle–chromosome complex. Scale bars = 10 μm. (**B**) Chromosome stiffness comparison between MI oocytes from 3- to 4-week-old mice ($n$ = 8) and 48-week-old mice ($n$ = 5). Young's modulus of 3- to 4-week-old MI oocyte chromosomes (3790 ± 700 Pa) is significantly lower than that of 48-week-old MI oocyte chromosomes (8150 ± 1590 Pa, p = 0.0150). Data are presented as mean ± SEM and statistical analysis was performed using $t$-test. *p < 0.05.

The online version of this article includes the following source data for figure 4:

**Source data 1.** Related to *Figure 4B*.

changes, contribute to the increased stiffness in older oocytes. Future investigations are needed to identify these age-related factors and their impact on chromosome mechanics.

## DNA damage reduces chromosome stiffness in oocytes

Oocytes from older individuals are known to exhibit higher levels of DNA damage compared to those from younger individuals (*Marangos et al., 2015*; *Horta et al., 2020*). In response to DNA damage on chromosomes, several DNA repair mechanisms are activated, which recruit various DNA repair proteins to the damage sites (*Marcon and Moens, 2005*). We hypothesized that this recruitment could affect chromosome stiffness. To test this hypothesis, we used etoposide, a chemotherapy drug used to treat a variety of cancers, including testicular and ovarian cancer (*Hoskins and Swenerton, 1994*; *Hainsworth and Greco, 1995*). We treated the oocytes with etoposide to introduce DNA damage and investigated its impact on chromosome stiffness (*Marangos et al., 2015*).

We cultured oocytes from the GV (germinal vesicle) stage for 6 hr to MI stage in the presence of a high concentration of etoposide (50 µg/ml). Following treatment, the spindle was isolated from the oocytes. Notably, the etoposide-treated chromosomes were unevenly distributed, forming large clusters in the spindle, unlike the well-aligned chromosomes in the control group (*Figure 5A*). 4′,6-Diamidine-2′-phenylindole dihydrochloride (DAPI) staining further confirmed that the etoposide-treated oocytes exhibited disrupted chromosome compaction and alignment compared to the control group (*Figure 5B*).

Upon measuring the stiffness of the chromosomes, we found that those from 50 µg/ml etoposide-treated MI oocytes were significantly less stiff than those from untreated control oocytes (1710 ± 430 vs. 3780 ± 700 Pa, p = 0.0245) (*Figure 5C*). Results at lower etoposide concentrations revealed that chromosome stiffness in untreated control oocytes was not significantly different from that in oocytes treated with 5 µg/ml etoposide (3780 ± 700 vs. 3930 ± 400 Pa, p = 0.8624). However, chromosome stiffness in untreated oocytes was significantly higher than that in oocytes treated with 25 µg/ml etoposide (3780 ± 700 vs. 1640 ± 340 Pa, p = 0.015) (*Figure 5C*).

Overall, these findings suggest that DNA damage reduces chromosome stiffness in oocytes, which is in accord with studies showing that DNA damage can make chromosomes softer (*dos Santos et al., 2021*). These results suggest that the increased chromosome stiffness observed in aged oocytes is not due to DNA damage.

## Discussion

In this study, we aspired to identify how meiotic cell cycle stage and aging influence meiotic chromosome stiffness. Prior studies found a large difference between mitotic and spermatocyte meiotic chromosome stiffness (*Biggs et al., 2020*). Spermatocyte meiotic prophase chromosomes were found to be 10 times stiffer than somatic chromosomes, which may be due to chromosome compaction or folding mechanisms specific to meiosis. During prophase I of meiosis, the SC zips up the two homologous chromosomes with components that are meiosis specific (*Gordon et al., 2021*). Previously, we found that SYCP1, an essential component of the SC that connects the lateral and central elements, does not contribute to the high prophase I chromosome stiffness in spermatocytes (*Biggs et al., 2020*), consistent with a 'liquid crystal' organizational scheme of the SC (*Rog et al., 2017*).

Here, we found that high chromosome stiffness also occurs in MI oocytes, which have a Young's modulus approximately 10 times larger than that of mitotic chromosomes, five times larger than that of MII chromosomes, and approximately the same as that of prophase I chromosomes (*Figure 2*). We note that the Young's modulus corrects for the varying thickness of chromosomes and indicates the elasticity in a volume- and geometry-independent way. However, the same trend of a large increase in chromosome stiffness during progression from somatic metaphase to meiotic prophase I and persisting through MI, followed by a reduction in chromosome stiffness upon progression to MII, is reflected in the force needed to double the length of chromosomes (*Figure 2—figure supplement 1*). This robust trend of a strong stiffening of chromosomes during prophase I and MI is in general accord with expectations from the chromosome compaction/relaxation and mechanics progression proposed by *Kleckner et al., 2004*. To further test Kleckner et al.'s model it would be desirable to measure the elasticity of somatic mitotic prophase chromosomes, but this has proven to be technically problematic in our attempts to date.

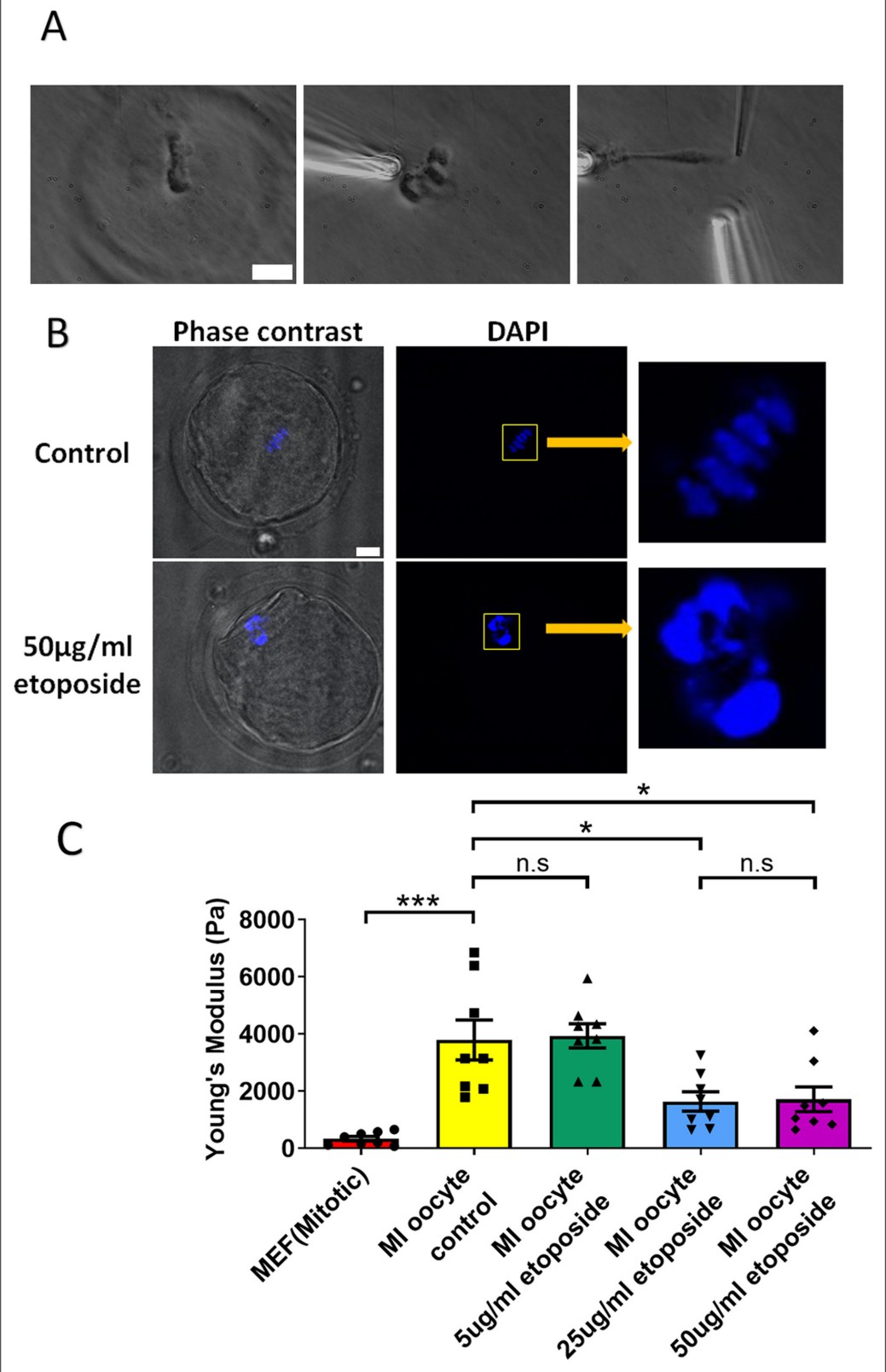

**Figure 5.** Etoposide treatment reduces chromosome stiffness. (**A**) Images of chromosome isolation from metaphase I (MI) oocyte treated with 50 μg/ml etoposide. Left panel: a spindle after cell lysis. Middle panel: a spindle captured with pipettes. Right panel: chromosome isolation. Scale bar = 10 μm. (**B**) 4′,6-Diamidine-2′-phenylindole dihydrochloride (DAPI) staining of control and 50 μg/ml etoposide-treated MI oocytes. Scale bar

*Figure 5 continued on next page*

*Figure 5 continued*

= 10 µm. (**C**) Chromosome stiffness comparison between mitotic cells (*n* = 8), control MI oocytes (*n* = 8), 5 µg/ml etoposide-treated MI oocytes (*n* = 8), 25 µg/ml etoposide-treated MI oocytes (*n* = 8), and 50 µg/ml etoposide-treated MI oocyte (*n* = 8). Young's modulus of control MI oocyte chromosomes (3790 ± 700 Pa) did not differ significantly from that of 5 µg/ml etoposide-treated MI oocyte chromosomes (3930 ± 400 Pa, p = 0.8624). However, it was significantly higher than that of 25 µg/ml etoposide-treated MI oocyte chromosomes (1640 ± 340 Pa, p = 0.015) and 50 µg/ml etoposide-treated MI oocyte chromosomes (1710 ± 430 Pa, p = 0.0245). Data are presented as mean ± SEM, with statistical analysis conducted using *t*-test. n.s, non-significant, *p < 0.05 and ***p < 0.001.

The online version of this article includes the following source data for figure 5:

**Source data 1.** Related to *Figure 5C*.

---

The SC central element dissociates from chromosomes before MI, in accord with our prior result that SYCP1 does not impart high meiotic chromosome stiffness (*Biggs et al., 2020*). We also measured chromosome stiffness in MII oocytes and found that it is significantly lower than in MI oocytes (in quantitative agreement with prior work *Hornick et al., 2015*, which measured a doubling force of MII chromosomes approximately double that of somatic mitotic chromosomes). Given the high and comparable stiffnesses of prophase I and MI chromosomes, we focused on factors specific to meiosis I, noting that AEs of the SC can still load onto chromosomes even in *Sycp1*$^{-/-}$ spermatocytes which display WT prophase I chromosome elasticity (*Biggs et al., 2020*).

Cohesin proteins are components of the meiotic prophase chromosome axis, and load onto chromosomes during DNA replication. Some meiosis-specific cohesins, such as REC8, persist on chromosome arms until anaphase I. We hypothesized that cohesins rather than the central element might contribute to meiotic chromosome stiffness, at least through MI. However, comparing results from experiments on WT and cohesin-deficient (*Rec8*$^{-/-}$, *Rad21l*$^{-/-}$, and *Stag3*$^{-/-}$) spermatocytes, we found no difference in chromosome stiffness. We conclude that the meiosis-specific cohesins do not account for high stiffness in meiotic chromosomes. While we do not yet understand the molecular origin of the high stiffness of meiotic chromosomes, we hypothesize that factors present during prophase I and persisting through MI, drive high meiotic chromosome stiffness.

A limitation of the current experiments is that we were restricted to analyzing homozygous meiotic cohesin mutants in prophase I in spermatocytes. While it would be valuable to compare these results with prophase I oocytes, this is currently not a practical option since oocytes reach this stage in utero, making genotyping at that time highly challenging. It would have been preferable to carry out control experiments on heterozygous spermatocytes, but the quantitative lack of any difference in prophase I mechanics across the four genetic cases studied (*Figure 3*) strongly suggests that cohesin mutations do not substantially affect chromosome mechanics at prophase I.

Another limitation is that our mechanics experiments are extracellular, following removal of chromosomes from MI and MII oocytes, prophase I spermatocytes and cultured somatic cells. The extracellular environment (phosphate-buffered saline [PBS], or cell culture buffer for somatic cells) is different from the environment inside the cell, and one might hypothesize that this changes chromosome mechanics, for example, via loss of chromosome-folding proteins after their isolation. In addition to past experiments indicating that mitotic chromosomes are stable for long periods after their isolation (*Pope et al., 2006*), we carried out control experiments on mouse oocyte chromosomes where we incubated them for 1 hr in PBS, or exposed them to a flow of Triton X-100 solution for 10 min; there was no change in chromosome stiffness in either case (Methods and *Figure 1—figure supplement 2*). Across all our experiments we found quantitative agreement across trials under corresponding conditions, arguing against there being large uncontrolled changes in chromosome structure resulting from our isolation method.

Previous research on mitotic chromosome stiffness revealed that chromosomes have a chromatin network organization (*Poirier and Marko, 2002*). Several factors, including condensin, have been found to affect chromosome stiffness (*Sun et al., 2018*). Condensin exists in two distinct complexes, condensin I and condensin II, and both are active during meiosis. Published studies indicate that condensin II is more sharply defined and more closely associated with the chromosome axis from anaphase I to metaphase II (*Lee et al., 2011*). Additionally, condensin II appears to play a more significant role in mitotic chromosome mechanics compared to condensin I (*Sun et al., 2018*). Thus, condensin II likely contributes more significantly to meiotic chromosome stiffness than condensin I.

It would be interesting to determine to what extent condensin defects affect meiotic chromosome structure.

Age also plays a role in altering chromosome stiffness. We found that chromosomes from aged MI oocytes had higher stiffness compared to those from younger oocytes, indicating that aging influences chromosome stiffness, in accord with a corresponding result for MII chromosomes (*Hornick et al., 2015*). This also provided further support for our conclusion that cohesins are not the primary factor making meiotic chromosomes stiffer as cohesin-protein levels reduce with age. The question of the molecular-level cause of oocyte meiotic chromosome stiffening with age remains open.

Hundreds of DNA double-stranded breaks are spontaneously introduced by the SPO11 protein at the onset of meiosis and aging also induces DNA damage (*Carofiglio et al., 2013*; *Schumacher et al., 2021*). How can meiotic chromosomes still become stiffer even in the presence of hundreds of DNA breaks in vivo? The answer may lie in the DNA repair proteins that are recruited to or near the DNA damage sites to repair the DNA damage and maintain genome integrity. Therefore, we hypothesized that DNA repair proteins contribute to meiotic chromosome stiffness. To test this hypothesis, we used a chemotherapy drug-etoposide to induce DNA damage in MI oocytes. Etoposide can increase the levels of TOP2–DNA covalent complexes, which generate DNA damage (*Menendez et al., 2022*). We found that etoposide treatment caused chromosome stiffness to decrease. This result is consistent with our previous study that DNA breaks can reduce chromosome stiffness in vitro (*Poirier and Marko, 2002*), reaffirming that meiotic chromosomes integrity relies on DNA continuity rather than the linkage of chromosome axis components (*Biggs et al., 2019*).

Since DNA repair proteins do not increase chromosome stiffness, other factors must be responsible for the high level of meiotic chromosome stiffness. During senescence, the amount of nuclear protein increases by around twofold, even though the cohesin level is decreased (*Tsutsumi et al., 2014*; *De Cecco et al., 2011*). Thus, it is possible that some nuclear proteins increase chromosome stiffness with age. However, further investigation is needed to determine which nuclear proteins contribute to chromosome stiffness. Additionally, histone methylation might influence chromosome stiffness because the level of histone methylation is proportional to chromosome stiffness (*Biggs et al., 2019*). Given that histone methylation levels are altered during aging, it is also plausible that some histone methyltransferases and demethylases regulate chromosome stiffness (*Soriano-Tárraga et al., 2019*; *Wang et al., 2022*). Moreover, histone methylation, especially H3K4, H3K9, and H3K36, plays a very important role in synapsis and recombination and has been found on meiotic chromatin from leptotene stage onward (*Wang et al., 2017*; *Hochwagen and Marais, 2010*; *Powers et al., 2016*). Further experiments could test the hypothesis that histone methylation regulates meiotic chromosome stiffness across different cell types.

No matter what factors are involved in stiffening meiotic chromosomes, they must alter chromosome organization to regulate chromosome stiffness. This indicates that chromosome stiffness is an important parameter for understanding chromosome structure. Defective chromosome organization is often related to various diseases, such as cancer, infertility, and senescence (*Thompson and Compton, 2011*; *Harton and Tempest, 2012*; *He et al., 2018*), and can be expected to cause changes in chromosome mechanics. By using micromanipulation experiments to study chromosome stiffness, we can better understand the mechanisms underlying these chromosome defects, potentially leading to new treatments and therapies.

A well-documented characteristic of aged oocytes is their higher rate of aneuploidy compared to that of younger oocytes (*Selesniemi et al., 2011*; *Ma et al., 2020*). The majority of aneuploidies can be traced to meiosis I, especially anaphase I, because it is an error-prone process (*Kolano et al., 2012*). It would be intriguing to investigate whether the increased chromosome stiffness contributes to this issue. A common cause of aneuploidy is lagging chromosomes during anaphase I (*Godek and Compton, 2018*). Chromosome stiffness may be related to the occurrence of lagging chromosomes, as chromosome separation relies on the pulling forces exerted by microtubules (*Duro and Marston, 2015*; *Cohen-Fix, 2000*). During anaphase I, homologous chromosomes, rather than sister chromatids, segregate. Thus, the chromosome region between the centromere and crossover must resist the tension applied by the spindle (*Duro and Marston, 2015*). Chromosomes must be stiff enough to prevent breakage under the pulling force. If the chromosomes are excessively stiff, the tightly connected homologous chromosomes may not properly separate, potentially causing aneuploidy. This suggests that chromosome stiffness change may be related to the high rate of aneuploidy observed

in aged oocytes. Modulating factors altering chromosome stiffness may help reduce the incidence of aneuploidy. Moreover, further investigation into the relationship between chromosome stiffness and lagging chromosomes could provide insight into the mechanisms underlying aneuploidy in oocytes.

## Materials and methods

**Key resources table**

| Reagent type (species) or resource | Designation | Source or reference | Identifiers | Additional information |
|---|---|---|---|---|
| Strain, strain background (*M. musculus*) | CD-1 | Charles River Laboratories, Wilmington, MA | RRID:IMSR_CRL:022 | |
| Strain, strain background (*M. musculus*) | C57BL/6 | This paper and papers from Dr. Philip W. Jordan's lab | RRID:IMSR_JAX:000664 | Mutant mouse lines maintained in Dr. Jordan's lab |
| Cell line (*M. musculus*) | Mouse embryonic fibroblasts | Dr. John Marko's lab | ATCC | |
| Chemical compound, drug | Etoposide | Cayman Chemical | Item No. 12092 | 50 µg/ml |
| Chemical compound, drug | M2 medium | Sigma-Aldrich | M7167-100ML | |
| Chemical compound, drug | M16 medium | Sigma-Aldrich | M7292-50ML | |
| Chemical compound, drug | EmbryoMax Acidic Tyrodes Solution | Sigma-Aldrich | MR-004-D | |
| Chemical compound, drug | IBMX | Sigma-Aldrich | I5879 | 100 µM |
| Chemical compound, drug | Triton X-100 | US Biological | 9002-93-1 | 0.05% |
| Chemical compound, drug | DAPI | Sigma-Aldrich | D9542 | 1:10,000 |
| Software, algorithm | LabVIEW | | RRID:SCR_014325 | |
| Software, algorithm | ImageJ | | RRID:SCR_003070 | |
| Software, algorithm | NIS-Elements | | RRID:SCR_014329 | |
| Software, algorithm | GraphPad Prism | | RRID:SCR_002798 | |

### Animals

WT CD-1 mice (Charles River Laboratories, Wilmington, MA) were used for all chromosome measurements, except for *Rec8*, *Stag3*, and *Rad21l* mutants, which were maintained on a C57BL/6 background. All mice were housed in the Pancoe CCM rooms of Northwestern University under a 12-hr dark/light cycle at 22 ± 1°C, with unrestricted access to food and water. All animal handling and experimental procedures were approved by the Institutional Animal Care and Use Committee (IACUC) at Northwestern University.

### Mouse oocyte in vitro culture

Oocytes were harvested from 3- to 4-week-old mice (48-week-old mice for aging studies) and culture in vitro. Both ovaries were immediately dissected and washed with M2 medium. The ovaries were then placed in prewarmed M2 medium containing 100 µM IBMX at 37°C. Sterilized needles were used to release cumulus–oocyte complexes (COCs). Next, COCs were pipetted in and out several times with a mouth pipette under dissection microscope to remove the cumulus cells surrounding the oocyte, yielding denuded oocytes at the GV stage. These oocytes were then placed in prewarmed M2 medium containing 100 µM IBMX.

Oocytes with irregular shapes and abnormal sizes were discarded, while healthy oocytes were selected and washed three times in M16 medium. The oocytes were then transferred to small drops of M16 medium covered with mineral oil in a petri dish and incubated at 37°C in a 5% $CO_2$ incubator. Depending on the experimental plan, oocytes were cultured with or without additional chemicals for 6 hr to reach the MI stage or 14 hr to reach the MII stage. After culture, the oocytes were rinsed three times with M2 medium and transferred to Tyrode's solution to remove the zona pellucida. Finally, the oocytes were transferred to PBS solution for chromosome stiffness measurements.

## Mitotic cell culture

MEFs were used for mitotic chromosome measurements. The MEF cells were cultured in DMEM (Corning) supplemented with 10% fetal bovine serum (HyClone) and 1% penicillin–streptomycin (100×, Corning). Cultures were maintained at 37°C in a 5% $CO_2$ incubator and passaged every 3–5 days, with a maximum of 20 passages.

For chromosome measurements, cells were transferred to prepared culture wells, which were made by fixing rubber rings on coverslips using wax. Each well was filled with 2 ml of culture media. The cells were cultured in these wells for 1–3 days to allow attachment to the coverslips. Chromosome measurements were conducted directly within these culture wells.

## Spermatocyte preparation

Testes were dissected from adult mice (*Biggs et al., 2020*). After removing the tunica albuginea, small clusters of seminiferous tubules were isolated and rinsed in PBS solution. These tubules were then finely chopped with a surgical blade to release spermatocytes, which were transferred into a culture well containing 2 ml of PBS. The spermatocytes were allowed to settle at the bottom of the well and subsequently used for chromosome measurements.

## Chromosome isolation

Hold pipettes, force pipettes, and stiff pipettes were prepared for manipulating chromosomes. They were made using a micropipette puller (Sutter P-97) and cut to appropriate sizes (*Biggs et al., 2020*). Chromosomes were isolated and measured under an inverted microscope (IX-70; Olympus) with a ×60 1.42 NA oil immersion objective and a ×1.5 magnification pullout. All experiments were conducted at room temperature (RT) within 3 hr to minimize the effects of water evaporation from the culture well.

MEF cells and spermatocytes were visually identified under phase-contrast microscopy. These cells were lysed with 0.05% Triton X-100 in PBS, applied using a spray pipette to remove cell membranes. After lysis, meiotic or mitotic chromosome bundles were released and captured by a pipette.

To isolate individual chromosomes, a force pipette was used to attach and extract a single chromosome from meiotic or mitotic chromosome bundles. Then, a stiff pipette was used to grab the other end of the chromosome, while the remaining chromosome bundles were carefully moved away.

For oocytes, a notable difference was that the spindle, along with its chromosomes, could be isolated as a unit. In this case, a force pipette was inserted into the spindle to capture chromosomes and drag them out from the spindle. After isolation, the chromosomes were ready for measurements.

## Chromosome stiffness measurement and calculation

Once the chromosome was held between the floppy force and moving stiff pipettes, it was stretched by moving the stiff pipettes perpendicularly. The process was recorded by LabVIEW software (*Biggs et al., 2020*). Before stretching, an image of the chromosome was captured to calculate the deflection of the force pipette. The stiff pipette was moved around 6.0 µm and then returned to its original position at a constant rate of 0.20 µm/s, with 0.04 µm steps controlled by the LabVIEW program. This measurement process was repeated six times.

The position of the stiff and force pipettes were recorded during the experiment. From the captured image, the original length and radius ($r$) of the chromosome were measured using ImageJ. The cross-section area ($A$) was calculated using the formula $A = \pi r^2/2$. The force constant of the force pipette was calibrated using a reference pipette with a premeasured spring constant (*Biggs et al., 2020*). Young's modulus ($E$) was calculated according to the formula $E = (F/A)/(\Delta L/L_0)$. $E$ was the Young's modulus, $F$ was the force applied to stretch the chromosome, $A$ was the cross-section area of the chromosome, $\Delta L$ was changed in chromosome length during stretching, and $L_0$ was the original length of the chromosome.

All measurements of meiotic chromosome stiffness were conducted in PBS solution. To investigate the effects of PBS exposure on chromosome stiffness, the Young's modulus of spermatocyte chromosomes was measured and compared before and after 1 hr of incubation in PBS (*Figure 1—figure supplement 2*, left panel). Similarly, to assess the impact of Triton X-100, the Young's modulus of spermatocyte chromosomes was measured and compared before and after microspraying with 0.05% Triton X-100 for 10 min (*Figure 1—figure supplement 2*, right panel). We observed no significant

change of chromosome stiffness in either of these cases, leading us to the conclusion that chromosome stiffness is not strongly affected by exposure to PBS or to Triton X-100.

## Chromosome staining

Isolated oocytes were fixed in 4% (wt/vol) paraformaldehyde in PBS solution for 30 min at RT. The oocytes were then washed three times with a washing buffer (0.1% Tween-20 and 0.01% Triton X-100 in PBS). Next, the oocytes were permeabilized in PBS containing 0.5% Triton X-100 for 20 min at RT. After that, the oocytes were transferred into 3% bovine serum albumin for blocking for 1 hr at RT. After blocking, the oocytes were washed three times and counterstained with 1 µg/ml of DAPI for 10 min at RT. Finally, the oocytes were washed twice with the washing buffer, mounted on glass slides in 80% glycerol, and examined using a Nikon A1R confocal microscope. Images were processed with NIS-Elements software.

## Acknowledgements

We thank all members of the Qiao and Marko groups for their technical support and valuable feedback on this manuscript. This work was supported by National Institutions of Health (NIH): R00 HD082375 and R01 GM135549, UM1 HG011536, R01 GM105847, R01 GM117155, T32 ES007326, and U54 CA203000.

## Additional information

### Competing interests

Philip W Jordan: is on the scientific advisory board of Gameto, Inc. The opinions and assertions expressed herein are those of the author(s) and do not reflect the official policy or position of the Uniformed Services University of the Health Sciences or the Department of Defense. The other authors declare that no competing interests exist.

### Funding

| Funder | Grant reference number | Author |
| --- | --- | --- |
| National Institutes of Health | R00 HD082375 | Huanyu Qiao |
| National Institutes of Health | R01 GM135549 | Huanyu Qiao |
| National Institutes of Health | UM1 HG011536 | John F Marko |
| National Institutes of Health | R01 GM105847 | John F Marko |
| National Institutes of Health | R01 GM117155 | Philip W Jordan |
| National Institutes of Health | T32 ES007326 | Huanyu Qiao |
| National Institutes of Health | U54 CA203000 | Wenan Qiang |

The funders had no role in study design, data collection, and interpretation, or the decision to submit the work for publication.

### Author contributions

Ning Liu, Investigation, Methodology, Validation, Visualization, Writing – original draft; Wenan Qiang, Investigation, Methodology, Resources; Philip W Jordan, Methodology, Resources, Writing – review and editing; John F Marko, Funding acquisition, Investigation, Methodology, Resources, Supervision, Validation, Writing – review and editing; Huanyu Qiao, Conceptualization, Funding acquisition,

Investigation, Methodology, Project administration, Resources, Supervision, Validation, Visualization, Writing – original draft, Writing – review and editing

## Author ORCIDs
Ning Liu ⬥ https://orcid.org/0009-0002-3738-4004
Wenan Qiang ⬥ https://orcid.org/0000-0001-5068-7128
Philip W Jordan ⬥ https://orcid.org/0000-0003-4890-2647
John F Marko ⬥ https://orcid.org/0000-0003-4151-9530
Huanyu Qiao ⬥ https://orcid.org/0000-0003-0966-8077

## Ethics
All mice were housed in the Pancoe CCM rooms of Northwestern University under a 12-hr dark/light cycle at 22 ± 1°C, with unrestricted access to food and water. All animal handling and experimental procedures were approved by the Institutional Animal Care and Use Committee (IACUC) at Northwestern University.

Reviewer #1 (Public review): https://doi.org/10.7554/eLife.97403.3.sa1
Reviewer #2 (Public review): https://doi.org/10.7554/eLife.97403.3.sa2
Author response https://doi.org/10.7554/eLife.97403.3.sa3

---

# Additional files

## Supplementary files
MDAR checklist

## Data availability
All data generated or analyzed during this study are included in the manuscript and supporting files.

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
