## [Editor Report · eLife Assessment]

This **valuable** paper describes the stiffness of meiotic chromosomes in both oocytes and spermatocytes. The authors identify differences in stiffness between meiosis I and II chromosomes, as well as an age-dependent increase in stiffness in meiosis I (and meiosis II) chromosomes, results that are highly significant for the field of chromosome biology. The report is, however, mostly descriptive and the mechanisms underlying age-dependent changes in chromosome stiffness remain unclear. The evidence suggesting that changes in stiffness are independent of cohesin, which is known to deteriorate with age, is still **incomplete**.

---

## [Referee Report · Reviewer #1 (Public review)]

Summary:

By using the biophysical chromosome stretching, the authors measured the stiffness of chromosomes of mouse oocytes in meiosis I (MI) and meiosis II (MII). This study was the follow-up of previous studies in spermatocytes (and oocytes) by the authors (Biggs et al. Commun. Biol. 2020: Hornick et al. J. Assist. Rep. and Genet. 2015). They showed that MI chromosomes are much stiffer (~10 fold) than mitotic chromosomes of mouse embryonic fibroblast (MEF) cells. MII chromosomes are also stiffer than the mitotic chromosomes. The authors also found that oocyte aging increases the stiffness of the chromosomes. Surprisingly, the stiffness of meiotic chromosomes is independent of meiotic chromosome components, Rec8, Stag3, and Rad21L. and aging increases the stiffness.

Strengths

This provides a new insight into the biophysical property of meiotic chromosomes, that is chromosome stiffness. The stiffness of chromosomes in meiosis prophase I is ~10-fold higher than that of mitotic chromosomes, which is independent of meiotic cohesin. The increased stiffness during oocyte aging is a novel finding.

Weaknesses:

A major weakness of this paper is that it does not provide any molecular mechanism underlying the difference between MI and MII chromosomes (and/or prophase I and mitotic chromosomes).

Comments on revisions:

The main text lacks the first page with the authors' names and their affiliations (and corresponding authors etc).

---

## [Referee Report · Reviewer #2 (Public review)]

Initial Review:

This paper reports investigations of chromosome stiffness in oocytes and spermatocytes> the paper shows that prophase I spermatocytes and MI/MII oocytes yield high Young Modulus values in the assay the authors applied. Deficiency in each one of three meiosis-specific cohesins they claim did not affect this result and increased stiffness was seen in aged oocytes but not in oocytes treated with the DNA-damaging agent etoposide.

The paper reports some interesting observations which are in line with a report by the same authors of 2020 where increased stiffness of spermatocyte chromosomes was already shown. In that sense, it the current manuscript is an extension of that previous paper and thus novelty is somewhat limited. The paper is also largely descriptive as it does neither propose mechanism nor report factors that determine the chromosomal stiffness.

There are several points that need to be considered.

Limitations of the study and the conclusions are not discussed in "Discussion"; that's a significant gap. Even more so as the authors rely on just one experimental system for all their data - no independent verification - and that in vitro system may be prone to artefacts.

It is somewhat unfortunate that they jump between oocytes and spermatocytes to address the cohesin question. Prophase I (pachytene) spermatocytes chromosomes are not directly comparable to MI or MII oocyte chromosomes. In fact, the authors report Young Modulus values of 3700 for MI oocytes and only 2700 for spermatocyte prophase chromosomes, illustrating this difference. Why not using oocyte-specific cohesin deficiencies?

It remains unclear whether the treatment of oocytes with the detergent TritonX-100 affects the spindle and thus the chromosomes isolated directly from the Triton-lysed oocytes. In fact, it is rather likely that the detergent affects chromatin-associated proteins and thus structural features of the chromosomes.

Why did the authors use mouse strains of different genetic background, CD-1 and C57BL/6? That makes comparison difficult. Breeding of heterozygous cohesin mutants will yield the ideal controls, i.e. littermates.

How did the authors capture chromosome axes from STAG3-deficient spermatocytes which feature very little if any axes? How representative are those chromosomes that could be captured?

Line 135: that statement is not substantiated; better to show retraction data and full reversibility.

Line 144: the authors claim that the Young Modulus of MII oocytes is "slightly" higher than that of mitotic cells (MEFs). Well, "slightly" means it is rather similar and therefore the commonly used statement that MII is similar to mitosis is OK - contrary to the authors claim.

There are a lot of awkward sentences in this text. Some sentences lack words, are not sufficiently precise in wording and/or logic, and there are numerous typos. Some examples can be found in lines 89 (grammar), 94, 95 ("looked"), 98, 101 ("difference" - between what?), and some are commonplaces or superficial (lines 92/93, 120...,). Occasionally the present and past tense are mixed (e.g. in M&M). Thus the manuscript is quite badly written.

Comments on revisions:

In their revised paper, Liu et al have addressed a number of my concerns and thus the paper is clearly improved in several details, e.g. in showing a control for a potential effect of the detergent (new supplies. fig. 5). Other points were not sufficiently addressed though.

I remain sceptical about using mice of a substantially different genetic background (CD1) as controls in the analysis of the cohesin mutants (C57BL/6). The argument that C57BL/6 yield smaller litter size is, frankly, ridiculous. Hundreds of labs worldwide extensively and successfully work with C57BL/6. Further, the paper Liu et al. cite to argue that there are no (or minor) differences in chromosome structure (Biggs et al., 2020, which is from the same lab) of the two mouse strains deals with spermatocyte chromosomes only. Nothing there on oocyte chromosomes. And there is no direct comparison within the same experimental setting since in Biggs et al only C57BL/6 is used (sic!). Thus, this is not a convincing argument. It would also be reassuring to see an independent reference directly comparing different genetic backgrounds (authors may have a look at older papers of Pat Hunt/Terry Hassold where they may find some data). In my experience, differences in genetic background do play a very clear role in meiosis, e.g. in the timing of juvenile spermatogenesis, in the onset of puberty, in the kinetics of oocyte maturation, in the success of PBE, and in biophysical properties as seen in the stability of oocytes during experimental handling. In fact, the authors themselves indicate differences in reproduction by stating the low litter size of C57BL/6. Thus, I strongly advise carrying out at least a few key experiments using C57BL/6 control mice (which can very easily and cheaply be obtained from vendors; the authors have used C57BL/6 wt before - see their 2020 paper).

The answer to my question #5 is not really satisfactory. I asked specifically how the authors isolated the very small chromosomes from Stag3-/- spermatocytes, where the axes are almost non-existing. The authors refer to suppl. fig. 3, but that shows isolation from Rec8-/- spermatocytes, which still have nicely visible, well-formed, shortened axes. Suppl. fig. 4 shows this for Rad21l-/-. Why not show this for the Stag3-/-, which in this respect is the most critical and difficult, and specifically answer my question?

The overall criticism of the lack of conceptual novelty of the basic message of the paper and of very little if any insights into the mechanisms and factors determining the changes in chromosome stiffness remains.

---

## [Author Response]

The following is the authors’ response to the original reviews.

**Reviewer #1 (Public Review):**
Summary:By using the biophysical chromosome stretching, the authors measured the stiffness of chromosomes of mouse oocytes in meiosis I (MI) and meiosis II (MII). This study was the follow-up of previous studies in spermatocytes (and oocytes) by the authors (Biggs et al. Commun. Biol. 2020: Hornick et al. J. Assist. Rep. and Genet. 2015). They showed that MI chromosomes are much stiffer (~10 fold) than mitotic chromosomes of mouse embryonic fibroblast (MEF) cells. MII chromosomes are also stiffer than the mitotic chromosomes. The authors also found that oocyte aging increases the stiffness of the chromosomes. Surprisingly, the stiffness of meiotic chromosomes is independent of meiotic chromosome components, Rec8, Stag3, and Rad21L. with aging.Strengths:This provides a new insight into the biophysical property of meiotic chromosomes, that is chromosome stiffness. The stiffness of chromosomes in meiosis prophase I is ~10-fold higher than that of mitotic chromosomes, which is independent of meiotic cohesin. The increased stiffness during oocyte aging is a novel finding.Weaknesses:A major weakness of this paper is that it does not provide any molecular mechanism underlying the difference between MI and MII chromosomes (and/or prophase I and mitotic chromosomes).

We acknowledge that our study does not provide a comprehensive explanation for the stage-related alterations in chromosome stiffness; however, we believe that the observation of these changes is itself of broad interest. Initially, we hypothesized that DNA damage or depletion of meiosis-specific cohesin might contribute to the observed increase in chromosome stiffness. However, our experimental finding did not support these hypotheses, indicating that neither DNA damage nor cohesion depletion is responsible for the stiffness increase. The molecular basis underlying the stage-related stiffness increase remains elusive and requires exploration in future studies. In the Discussion, we propose that factors such as condensin, nuclear proteins, and histone methylation may play a role in regulating meiotic chromosome stiffness. The involvement of these factors in stage-related chromosome stiffening requires future investigation.

**Reviewer #2 (Public Review):**
This paper reports investigations of chromosome stiffness in oocytes and spermatocytes. The paper shows that prophase I spermatocytes and MI/MII oocytes yield high Young Modulus values in the assay the authors applied. Deficiency in each one of three meiosis-specific cohesins they claim did not affect this result and increased stiffness was seen in aged oocytes but not in oocytes treated with the DNA-damaging agent etoposide.The paper reports some interesting observations which are in line with a report by the same authors of 2020 where increased stiffness of spermatocyte chromosomes was already shown. In that sense, the current manuscript is an extension of that previous paper, and thus novelty is somewhat limited. The paper is also largely descriptive as it does neither propose a mechanism nor report factors that determine the chromosomal stiffness.There are several points that need to be considered.(1) Limitations of the study and the conclusions are not discussed in the "Discussion" section and that is a significant gap. Even more so as the authors rely on just one experimental system for all their data - there is no independent verification - and that in vitro system may be prone to artefacts.

Our experimental system has been used to study different types of chromosome stiffness as well as nuclear stiffness. We have compared our results with previously published data and found the data is consistent across different experiments. To address the reviewer’s concern, we describe the limitations of our in vitro experimental approach in the Discussion section.

(2) It is somewhat unfortunate that they jump between oocytes and spermatocytes to address the cohesin question. Prophase I (pachytene) spermatocytes chromosomes are not directly comparable to MI or MII oocyte chromosomes. In fact, the authors report Young Modulus values of 3700 for MI oocytes and only 2700 for spermatocyte prophase chromosomes, illustrating this difference. Why not use oocyte-specific cohesin deficiencies?

In this study, our goal was to investigate the mechanism underlying the increased chromosome stiffness observed during prophase I. Ideally, we would have compared wild-type and cohesin-deleted mouse oocytes at the metaphase I (MI) stage. However, experimental constraints made this approach unfeasible: spermatocytes and oocytes from *Rec8-/-* and *Stag3-/-* mutant mice cannot reach MI stage, and *Rad21l-/-* mutant mice are sterile in males and subfertile in females, because cohesin proteins are crucial for germline cell development.

Additionally, collecting prophase I chromosomes from oocytes is exceptionally challenging and requires fetal mice as prophase I oocyte sources because female oocytes progress to the diplotene stage during fetal development. The process is further complicated by the difficulty of genotyping fetal mice, making the study of female prophase I impracticable. By contrast, spermatocytes are continuously generated in males throughout life, with meiotic stages readily identifiable, making them more accessible for analysis.

Our findings consistently showed increased chromosome stiffness in both prophase I spermatocytes and MI oocytes, suggesting that the phenomenon is not sex-specific. This observation implies that similar effects on chromosome stiffness may occur across meiotic stages, from prophase I to MI.

(3) It remains unclear whether the treatment of oocytes with the detergent TritonX-100 affects the spindle and thus the chromosomes isolated directly from the Triton-lysed oocytes. In fact, it is rather likely that the detergent affects chromatin-associated proteins and thus structural features of the chromosomes.

Regarding the use of Triton X-100, it is important to emphasize that the concentration used (0.05%) is very low and unlikely to significantly affect chromosome stiffness. To support this assertion, we have provided additional evidence in the revised manuscript demonstrating that this low concentration of Triton X-100 has a negligible effect on chromosome stiffness (Supplement Fig. 5, Right panel).

(4) Why did the authors use mouse strains of different genetic backgrounds, CD-1, and C57BL/6? That makes comparison difficult. Breeding of heterozygous cohesin mutants will yield the ideal controls, i.e. littermates.

The genetic mutant mice, all in a C57BL/6 background, were generously provided by Dr. Philip Jordan and delivered to our lab. As our lab does not currently maintain C57BL/6 colony and given that this strain typically produces small litter sizes - which would have complicated the remainder of the study - we chose CD-1 mice as the control group and used C57BL/6 mice specifically for the cohesin study. To address potential concerns regarding genetic background differences, we compared our results with previously published data from C57BL/6 mice and found no significant differences (2710 ± 610 Pa versus 3670 ± 840 Pa, P = 0.4809) (Biggs et al., 2020). Furthermore, prophase I spermatocytes from CD-1 mice showed no significant difference compared to any of the three cohesin-deleted C57BL/6 mutant mice, suggesting that chromosome stiffness is not significantly influenced by genetic background.

(5) How did the authors capture chromosome axes from STAG3-deficienct spermatocytes which feature very few if any axes? How representative are those chromosomes that could be captured?

We isolated chromosomes from prophase I mutant spermatocytes, which were identified by their large size, round shape, and thick chromosomal threads - characteristics indicative of advanced condensation and a zygotene-like stage during prophase I (Supplemental Fig. 3). The methodology for isolating these chromosomes has been described in details in our previous publication (Biggs et al., 2020), which is referenced in the current manuscript.

**Reviewer #3 (Public Review):**
Summary:Understanding the mechanical properties of chromosomes remains an important issue in cell biology. Measuring chromosome stiffness can provide valuable insights into chromosome organization and function. Using a sophisticated micromanipulation system, Liu et al. analyzed chromosome stiffness in MI and MII oocytes. The authors found that chromosomes in MI oocytes were ten-fold stiffer than mitotic ones. The stiffness of chromosomes in MI mouse oocytes was significantly higher than that in MII oocytes. Furthermore, the knockout of the meiosis-specific cohesin component (Rec8, Stag3, Rad21l) did not affect meiotic chromosome stiffness. Interestingly, the authors showed that chromosomes from old MI oocytes had higher stiffness than those from young MI oocytes. The authors claimed this effect was not due to the accumulated DNA damage during the aging process because induced DNA damage reduced chromosome stiffness in oocytes.Strengths:The technique used (isolating the chromosomes in meiosis and measuring their stiffness) is the authors' specialty. The results are intriguing and informative to the chromatin/chromosome and other related fields.Weaknesses:(1) How intact the measured chromosomes were is unclear.

Currently, a well-calibrated chromosome mechanics experiment requires the extracellular isolation of chromosomes. In experiments conducted parallel to those in our previous study (Biggs et al., 2020), we obtained quantitatively consistent results, including measurements of the Young modulus for prophase I spermatocyte chromosomes. Our isolation approach is significantly gentler than bulk methods that rely on hypotonic buffer-driven cell lysis and centrifugation. If substantial chromosomal damage had occurred during isolation, we would expect greater variation between experiments, as different amounts or types of damage could influence the results.

(2) Some control data needs to be included.

We used wild-type prophase I spermatocytes and metaphase I (MI) oocytes as controls. To validate our findings, we compared some of our results with those reported in a previous study and observed consistent outcomes (Biggs et al., 2020).

(3) The paper was not well-written, particularly the Introduction section.

We have revised the paper and improved the overall quality of the manuscript.

(4) How intact were the measured chromosomes? Although the structural preservation of the chromosomes is essential for this kind of measurement, the meiotic chromosomes were isolated in PBS with Triton X-100 and measured at room temperature. It is known that chromosomes are very sensitive to cation concentrations and macromolecular crowding in the environment (PMID: 29358072, 22540018, 37986866). It would be better to discuss this point.

As suggested, we investigated the impact of PBS and Triton X-100 on chromosome stiffness. Our findings indicate that neither PBS nor Triton X-100 caused significant changes in chromosome stiffness (Supplemental Fig. 5).

**Recommendations For The Authors:**

Major points of Reviewers that the Editor indicated should be addressed

(1) Reviewer's point 3, the effect of the high concentration of etoposide: It would be advisable to use lower concentrations of etoposide to observe the effect of DNA damage on chromosome stiffness more accurately.

The effect of etoposide on oocyte is dose-dependent (Collins et al., 2015). Oocytes are generally not highly sensitive to DNA damage, and even at relatively high concentrations, not all may exhibit a response. To ensure that sufficient DNA damage in the oocytes we isolated, we used relatively high concentration of etoposide for the experiment. This concentration (50 μg/ml) falls within the typical range reported in the literature (Marangos and Carroll, 2012)(Cai et al., 2023)(Lee et al., 2023). As the reviewer suggested, we tested two additional lower concentrations of etoposide (5 μg/ml and 25 μg/ml) (see Fig. 5 C). We did not observe any significant differences in chromosome stiffness in 5 µg/ml etoposide-treated oocytes compared to the control. However, higher concentrations of etoposide (25 μg/ml) significantly reduced oocyte chromosome stiffness compared to the control.

Revision to manuscript:

“Results at lower etoposide concentrations revealed that chromosome stiffness in untreated control oocytes was not significantly different from that in oocytes treated with 5 μg/ml etoposide (3780 ± 700 Pa versus 3930 ± 400 Pa, P = 0.8624). However, chromosome stiffness in untreated oocytes was significantly higher than that in oocytes treated with 25 μg/ml etoposide (3780 ± 700 Pa versus 1640 ± 340 Pa, P = 0.015) (Figure 5C).”

(2) Reviewer's point 3, the effect of Triton X-100: This is related to the concern of the #3 reviewer. It is critical to check whether the detergent does not affect the stiffness indirectly or not.

To demonstrate that the low concentration of Triton X-100 does not influence chromosome stiffness, we conducted additional experiments. First, we isolated chromosomes and measured their stiffness. Then, we treated the chromosomes with 0.05% Triton X-100 via micro-spraying and remeasured the stiffness. The results showed no significant difference (see Supplement Fig. 5 right panel).

Revision to manuscript:

“In addition to past experiments indicating that mitotic chromosomes are stable for long periods after their isolation (Pope et al., 2006), we carried out control experiments on mouse oocyte chromosomes where we incubated them for 1 hour in PBS, or exposed them to a flow of Triton X-100 solution for 10 minutes; there was no change in chromosome stiffness in either case (Methods and Supplementary Fig. 5).”

(3) Reviewer's point 1, the effect of the buffer composition: Please describe how the composition affects the stiffness of the chromosomes.

PBS is an economical and effective buffer solution that closely mimics the osmotic conditions of the cytoplasm, which is crucial for maintaining chromosomal structural integrity. Appropriate ion concentrations are crucial for preserving chromosome integrity, as imbalances—either too high or too low—can alter chromosome morphology (Poirier and Marko, 2002). When chromosomes are stored in PBS, their stiffness remains relatively stable, even with prolonged exposure, ensuring minimal changes to their physical properties. To confirm this, we isolated chromosomes and measured their stiffness. After one-hour incubation in PBS, we remeasured stiffness and observed no significant differences, which demonstrated that chromosomes remain stable in PBS (see Supplement Fig.5 left panel).

Revision to manuscript:

“In this study, we developed a new way to isolate meiotic chromosomes and measure their stiffness. However, one concern is that the measurements were conducted in PBS solution, which is different from the intracellular environment. To address this, we monitored chromosome stiffness overtime in PBS solution and found that it remained stable over a period of one hour (Supplement Fig. 5 Left panel).”

**Reviewer #1 (Recommendations For The Authors):**
Major points:(1) Previously, the role of condensin complexes in chromosome stiffness is shown (Sun et al. Chromosome Research, 2018). Thus, at least the authors described the condensin staining on MI and MII chromosomes.

We have added sentences in the discussion to elaborate on the role of condensin.

Revision to manuscript:

“Several factors, including condensin, have been found to affect chromosome stiffness (Sun et al., 2018). Condensin exists in two distinct complexes, condensin I and condensin II, and both are active during meiosis. Published studies indicate that condensin II is more sharply defined and more closely associated with the chromosome axis from anaphase I to metaphase II (Lee et al., 2011). Additionally, condensin II appears to play a more significant role in mitotic chromosome mechanics compared to condensin I (Sun et al., 2018). Thus, condensin II likely contributes more significantly to meiotic chromosome stiffness than condensin I.”

(2) Although the authors nicely showed the difference in the stiffness between MI and MII chromosomes (Figure 2), as known, MI chromosomes are bivalent (with four chromatids) while MII chromosomes are univalent (with two chromatids). The physical property of the chromosomes would be affected by the number of chromatids. It would be essential for the authors to measure the physical properties of a univalent of MI chromosomes from mice defective in meiotic recombination such as Spo11 and/or Mlh3 KO mice.

The reviewer correctly pointed out that the number of chromatids in chromosomes differs between metaphase I (MI) and metaphase II (MII) stages. We have addressed this difference by calculating Young’s modulus (E), a mechanical property that describes the elasticity of a material, independent of its geometry. Young’s modulus describes the intrinsic properties of the material itself, rather than the specific characteristics of the object being tested. It is calculated as E=(F/A)/(∆L/L0), where F was the force given to stretch the chromosome, A was the cross-section area, ∆L was the length change of the chromosome, and L0 was the original length of the chromosome. While an increase in chromosome or chromatid numbers, results in a larger cross-sectional area, leading to a higher doubling force (F). This variation in chromosome number or cross-sectional area does not impact the calculation of chromosome stiffness/Young’s modulus (E). While study of the mutants suggested by the referee would certainly be interesting, it would be likely that the absence of these key recombination factors would impact chromosome stiffness in a more complex way than just changing their thickness; this type of study is beyond the scope of the present manuscript and is an exciting direction for future studies.

(3) In Figure 5, the authors measure the stiffness of etoposide-treated MI chromosomes. The concentration of the drug was 50 ug/ml, which is very high. The authors should analyze the different concentrations of the drug to check the chromosome stiffness. Moreover, etoposide is an inhibitor of Topoisomerase II. The effect of the drug might be caused by the defective Top2 activity, rather than Top2-adducts, thus DNA damage. It is very important to check the other Top2 inhibitors or DNA-damaging agents to generalize the effect of DNA damage on chromosome stiffness. Moreover, DNA damage induces the DNA damage response. It is important to check the effect of DDR inhibitors on the damage-induced change of stiffness.

The reviewer is correct in noting that etoposide can induce DNA damage and inhibit Top2 activity. To address this concern, our previous DNase experiment provided further clarity and supports our results of this study (Biggs et al., 2020). This experiment was conducted in vitro, where DNase treatment caused DNA damage on chromosomes without affecting Top2 activity or triggering DNA damage response. The results demonstrated that DNase treatment led to reduced chromosome stiffness, which aligns with the findings presented in our manuscript.

(4) In the same line as the #3 point, the authors also need to check the effect of etoposide on the stiffness of mitotic chromosomes from MEF.

Experiments on MEF mitotic chromosomes were designed to serve as a reference for the meiotic chromosome studies. The etoposide experiments on meiotic chromosomes specifically aimed to investigate how DNA damage affects meiotic chromosome structure. While it would be interesting to explore the effects of etoposide-induced DNA damage on mitotic chromosomes, it represents a distinct research question that falls outside the scope of the current study.

Minor points:(1) Line 141-142: Previous studies by the author analyzed the stiffness of mitotic chromosomes from pro-metaphase. Which stage of cell cycles did the authors analyze here?

To ensure consistency in our experiments, we also measured the stiffness of mitotic chromosomes at the prometaphase stage. The precise stage used is very near to metaphase, at the very end of the prometaphase stage. We have modified the manuscript to clarify this point.

Revision to manuscript:

“For comparison with the meiotic case, we measured the chromosome stiffness of Mouse Embryonic Fibroblasts (MEFs) at late pro-metaphase (just slightly before their attachment to the mitotic spindle) and found that the average Young’s modulus was 340 ± 80 Pa (Figure 2B). The value is consistent with our previously published data, where the modulus for MEFs was measured to be 370 ± 70 Pa (Biggs et al., 2020).”

(2) Line 157: Here, the doubling force of MI (and MII) oocytes should be described in addition to those of spermatocytes.

The purpose of this paragraph is to demonstrate the reproductivity and consistency of our experiments. In this section, we compared our data with previously published findings. Published data do not include chromosome stiffness measurement from MI mouse oocytes. Our experiment is the first to assess this. Therefore, we did not include MI mouse oocytes in that comparison. To clarify this, we have added sentences to highlight the comparison of doubling force.

Revision to manuscript:

“Here, we found that the doubling forces of chromosomes from MI and MII oocytes are 3770 ± 940 pN and 510 ± 50 pN, respectively. We conclude that chromosomes from MI oocytes are much stiffer than those from both mitotic cells and MII oocytes (Supplement Fig. 2), in terms of either Young’s modulus or doubling force.”

(3) Line 202: What stage of prophase I do the authors mean by the spermatocyte stage here? Diakinesis, Metaphase I or prometaphase I? I am not sure how the authors can determine a specific stage of prophase I by only looking at the thickness of the chromosomes. Please show the thickness distribution of WT and *Rec8-/-* chromosomes.

We have reworded the sentence and clarified that the spermatocyte stage is prophase I stage. Since *Rec8-/-* spermatocytes cannot progress beyond the pachytene stage of prophase I, the isolated chromosomes must be in prophase I rather than diakinesis, metaphase I, prometaphase I, or any later stages (Xu et al., 2005). Based on the cell size and degree of chromosome condensation (Biggs et al., 2020), it is most likely that the measured chromosomes are at the zygotene-like stage. However, as we cannot definitively determine the exact substage of prophase I, thus, we have referred to them simply as prophase I.

Revision to manuscript:

“We isolated chromosomes from *Rec8-/-* prophase I spermatocytes, which displayed large and round cell size and thick chromosomal threads, indicative of advanced chromosome compaction after stalling at a zygotene-like prophase I stage (Supplement Fig. 3). The combination of large cell size and degree of chromosome compaction allowed us to reliably identify *Rec8-/-* prophase I chromosomes. Using micromanipulation, we measured chromosome stiffness by stretching the chromosomes (Supplement Fig. 3) (Biggs et al., 2019).”

**Reviewer #2 (Recommendations For The Authors):**
(1) Line 135: that statement is not substantiated; better to show retraction data and full reversibility.

We added a figure showing oocyte chromosome stretching, which showed that the oocyte chromosome is elastic, and that the stretching process is reversible (Supplement Fig.1).

(2) Line 144: the authors claim that the Young Modulus of MII oocytes is "slightly" higher than that of mitotic cells (MEFs). Well, "slightly" means it is rather similar, and therefore the commonly used statement that MII is similar to mitosis is OK - contrary to the authors' claim.

We have removed the word “slightly” in the manuscript. The difference is statistically significant.

Revision to manuscript:

“Surprisingly, despite this reduction, the stiffness of MII oocyte chromosomes was still significantly higher than that for mitotic cells (Figure 2B).”

(3) There are a lot of awkward sentences in this text. Some sentences lack words, are not sufficiently precise in wording and/or logic, and there are numerous typos. Some examples can be found in lines 89 (grammar), 94, 95 ("looked"), 98, 101 ("difference" - between what?), and some are commonplaces or superficial (lines 92/93, 120...,). Occasionally the present and past tense are mixed (e.g. in M&M). Thus the manuscript is quite poorly written.

Thanks for the comments of the reviewer. We have revised all the sentences highlighted by the reviewer and polished the entire manuscript.

**Reviewer #3 (Recommendations For The Authors):**
(1) Line 48. "We then investigated the contribution of meiosis-specific cohesin complexes to chromosome stiffness in MI and MII oocytes." There is no data on oocytes with meiosis-specific cohesin KO. This part should be corrected.

We have corrected this error.

Revision to manuscript:

“We examined the role of meiosis-specific cohesin complexes in regulating chromosome stiffness.”

(2) Lines 155-157. The result of MI mouse oocyte chromosomes should also be mentioned here (Supplementary Figure 1).

Please see our response to Reviewer 1 – Minor Point 2.

(3) Line 163. "The stiffness of chromosomes in MI mouse oocytes is significantly higher compared to MII oocytes."Is this because two homologs are paired in MI chromosomes (but not in MII chromosomes)? The authors may want to discuss the possible mechanism.

Please see our response to Reviewer 1 – Major Point 2.

(4) Line 188: "We hypothesized that MI oocytes... would have higher chromosome stiffness than MII oocytes." Why did the authors measure chromosomes from spermatocytes but not MI oocytes?

Both spermatocytes and oocytes from *Rec8-/-*, *Stag3-/-*, and *Rad21l-/-* mutant mice cannot reach MI stage because cohesin proteins are crucial for germline-cell development. We chose to use spermatocytes in our study because collecting fetal meiotic oocytes is extremely difficult, and genotyping fetal mice adds another layer of complexity to the experiments. In females, all oocytes complete prophase I and progress to the dictyotene stage during the fetal stage. Obtaining individual oocytes at this stage is challenging. In contrast, spermatocytes are continuously generated at all stages in males.

(5) To support the authors' conclusion, verifying the KO of REC8, STAG3, and RAD21L by immunostaining or other methods is essential.

These mice are provided by one of the authors, Dr. Philip Jordan, who has published several papers using these knockout mice (Hopkins et al., 2014)(Ward et al., 2016). The immunostaining of these models has already been well-characterized in those previous studies. In addition to performing double genotyping, we also use the size of the collected testes as an additional verification of the mutant genotype. These knockout mice have significantly smaller testes compared to their wild-type counterparts, providing a clear physical indicator of the mutation.

(6) Some of the cited papers and descriptions in the Introduction are not appropriate and confusing. This part should be improved:

Line 79. Recent studies have revealed that the 30-nm fiber is not considered the basic structure of chromatin (e.g., review, PMID: 30908980; original papers, PMID: 19064912, 22343941, 28751582). This point should be included.

We have corrected the references as needed. Additionally, thank you for the updated information regarding the 30-nm fiber. We have removed all the descriptions about the 30-nm fiber to ensure the information is accurate and up to date.

(7) Line 83. Reviews on mitotic chromosomes, rather than Ref. 9, should be cited here. For instance, PMID: 33836947, 31230958.

We have corrected it and added references according to the review’s suggestion.

(8) Line 85. Refs. 10 and 11 are not on the "Scaffold/Radial-Loop" model. For instance, PMID: 922894, 277351, 12689587. The other popular model is the hierarchical helical folding model (PMID: 98280, 15353545).

We have corrected it and added appropriate references according to the review’s suggestion. Regarding the hierarchical helical folding model, our experiments do not provide data that either support or refute this model. Thus, we have opted not to include any discussion of this model in our manuscript.

(9) Figure legends. There is no description of the statistical test.

We have added the description of the statistical test at the end of the figure legends for clarity.

(10) Line 156. The authors should mention which stages in spermatocyte prophase I (pachytene?) were used for their measurement.

We cannot precisely determine the substage of prophase I in the spermatocytes although it is most likely in the pachytene stage.

(11) Line 241. "DNA damage reduces chromosome stiffness in oocytes." It would be better to show how much damage was induced in aged and etoposide-treated chromosomes, for example, by gamma-H2AX immunostaining. In addition, there are some papers that show DNA damage makes chromatin/chromosomes softer (e.g., PMID: 33330932). The authors need to cite these papers.

The effects of etoposide and age on meiotic oocytes has been published (Collins et al., 2015) (Marangos et al., 2015)(Winship et al., 2018).

We are grateful for the citation information provided by the reviewer and have added it to our manuscript.

Revision to manuscript:

“Overall, these findings suggest that DNA damage reduces chromosome stiffness in oocytes instead of increasing it, which aligns with other studies showing that DNA damage can make chromosomes softer (Dos Santos et al., 2021). These results suggest that the increased chromosome stiffness observed in aged oocytes is not due to DNA damage.”

(12) Line 328. Senescence?

This error is corrected in the revised manuscript.

Revision to manuscript:

“Defective chromosome organization is often related to various diseases, such as cancer, infertility, and senescence (Thompson and Compton, 2011; Harton and Tempest, 2012; He et al., 2018).”

References:

Biggs, R., P.Z. Liu, A.D. Stephens, and J.F. Marko. 2019. Effects of altering histone posttranslational modifications on mitotic chromosome structure and mechanics. *Mol. Biol. Cell*. 30:820–827. doi:10.1091/mbc.E18-09-0592.

Biggs, R.J., N. Liu, Y. Peng, J.F. Marko, and H. Qiao. 2020. Micromanipulation of prophase I chromosomes from mouse spermatocytes reveals high stiffness and gel-like chromatin organization. *Commun. Biol.* 3:1–7. doi:10.1038/s42003-020-01265-w.

Cai, X., J.M. Stringer, N. Zerafa, J. Carroll, and K.J. Hutt. 2023. Xrcc5/Ku80 is required for the repair of DNA damage in fully grown meiotically arrested mammalian oocytes. *Cell Death Dis.* 14:1–9. doi:10.1038/s41419-023-05886-x.

Collins, J.K., S.I.R. Lane, J.A. Merriman, and K.T. Jones. 2015. DNA damage induces a meiotic arrest in mouse oocytes mediated by the spindle assembly checkpoint. *Nat. Commun.* 6. doi:10.1038/ncomms9553.

Harton, G.L., and H.G. Tempest. 2012. Chromosomal disorders and male infertility. *Asian J. Androl.* 14:32–39. doi:10.1038/aja.2011.66.

He, Q., B. Au, M. Kulkarni, Y. Shen, K.J. Lim, J. Maimaiti, C.K. Wong, M.N.H. Luijten, H.C. Chong, E.H. Lim, G. Rancati, I. Sinha, Z. Fu, X. Wang, J.E. Connolly, and K.C. Crasta. 2018. Chromosomal instability-induced senescence potentiates cell non-autonomous tumourigenic effects. *Oncogenesis*. 7. doi:10.1038/s41389-018-0072-4.

Hopkins, J., G. Hwang, J. Jacob, N. Sapp, R. Bedigian, K. Oka, P. Overbeek, S. Murray, and P.W. Jordan. 2014. Meiosis-Specific Cohesin Component, Stag3 Is Essential for Maintaining Centromere Chromatid Cohesion, and Required for DNA Repair and Synapsis between Homologous Chromosomes. *PLoS Genet.* 10:e1004413. doi:10.1371/journal.pgen.1004413.

Lee, C., J. Leem, and J.S. Oh. 2023. Selective utilization of non-homologous end-joining and homologous recombination for DNA repair during meiotic maturation in mouse oocytes. *Cell Prolif.* 56:1–12. doi:10.1111/cpr.13384.

Lee, J., S. Ogushi, M. Saitou, and T. Hirano. 2011. Condensins I and II are essential for construction of bivalent chromosomes in mouse oocytes. *Mol. Biol. Cell*. 22:3465–3477. doi:10.1091/mbc.E11-05-0423.

Marangos, P., and J. Carroll. 2012. Oocytes progress beyond prophase in the presence of DNA damage. *Curr. Biol.* 22:989–994. doi:10.1016/j.cub.2012.03.063.

Marangos, P., M. Stevense, K. Niaka, M. Lagoudaki, I. Nabti, R. Jessberger, and J. Carroll. 2015. DNA damage-induced metaphase i arrest is mediated by the spindle assembly checkpoint and maternal age. *Nat. Commun.* 6:1–10. doi:10.1038/ncomms9706.

Poirier, M.G., and J.F. Marko. 2002. Mitotic chromosomes are chromatin networks without a mechanically contiguous protein scaffold. *Proc. Natl. Acad. Sci. U. S. A.* 99:15393–15397. doi:10.1073/pnas.232442599.

Pope, L.H., C. Xiong, and J.F. Marko. 2006. Proteolysis of Mitotic Chromosomes Induces Gradual and Anisotropic Decondensation Correlated with a Reduction of Elastic Modulus and Structural Sensitivity to Rarely Cutting Restriction Enzymes. *Mol. Biol. Cell*. 17:104. doi:10.1091/MBC.E05-04-0321.

Dos Santos, Á., A.W. Cook, R.E. Gough, M. Schilling, N.A. Olszok, I. Brown, L. Wang, J. Aaron, M.L. Martin-Fernandez, F. Rehfeldt, and C.P. Toseland. 2021. DNA damage alters nuclear mechanics through chromatin reorganization. *Nucleic Acids Res.* 49:340–353. doi:10.1093/nar/gkaa1202.

Sun, M., R. Biggs, J. Hornick, and J.F. Marko. 2018. Condensin controls mitotic chromosome stiffness and stability without forming a structurally contiguous scaffold. *Chromosom. Res.* 26:277–295. doi:10.1007/s10577-018-9584-1.

Thompson, S.L., and D.A. Compton. 2011. Chromosomes and cancer cells. *Chromosom. Res.* 19:433–444. doi:10.1007/s10577-010-9179-y.

Ward, A., J. Hopkins, M. Mckay, S. Murray, and P.W. Jordan. 2016. Genetic Interactions Between the Meiosis-Specific Cohesin Components, STAG3, REC8, and RAD21L. *G3 (Bethesda).* 6:1713–24. doi:10.1534/g3.116.029462.

Winship, A.L., J.M. Stringer, S.H. Liew, and K.J. Hutt. 2018. The importance of DNA repair for maintaining oocyte quality in response to anti-cancer treatments, environmental toxins and maternal ageing. *Hum. Reprod. Update*. 24:119–134. doi:10.1093/humupd/dmy002.

Xu, H., M.D. Beasley, W.D. Warren, G.T.J. van der Horst, and M.J. McKay. 2005. Absence of Mouse REC8 Cohesin Promotes Synapsis of Sister Chromatids in Meiosis. *Dev. Cell*. 8:949–961. doi:10.1016/j.devcel.2005.03.018.